# Genetic Background of Polycythemia Vera

**DOI:** 10.3390/genes13040637

**Published:** 2022-04-02

**Authors:** Mathilde Regimbeau, Romain Mary, François Hermetet, François Girodon

**Affiliations:** 1Inserm U1231, Université de Bourgogne, 21000 Dijon, France; mathilde.regimbeau@gmail.com (M.R.); romain.mary1@gmail.com (R.M.); francois.hermetet@u-bourgogne.fr (F.H.); 2Institut IMAGINE, Inserm U1163, 75015 Paris, France; 3Service d’Hématologie Biologique, Hôpital du Bocage, CHU de Dijon, 21000 Dijon, France; 4Laboratoire d’Excellence GR-Ex (Labex GR-Ex), 75015 Paris, France; 5Member of France Intergroupe Syndromes Myéloprolifératifs (FIM), 75010 Paris, France

**Keywords:** polycythemia vera, *JAK2* mutation, erythrocytosis, myeloproliferative neoplasm, mutational landscape

## Abstract

Polycythemia vera belongs to myeloproliferative neoplasms, essentially by affecting the erythroblastic lineage. JAK2 alterations have emerged as major driver mutations triggering PV-phenotype with the *V617F* mutation detected in nearly 98% of cases. That’s why JAK2 targeting therapeutic strategies have rapidly emerged to counter the aggravation of the disease. Over decades of research, to go further in the understanding of the disease and its evolution, a wide panel of genetic alterations affecting multiple genes has been highlighted. These are mainly involved in alternative splicing, epigenetic, miRNA regulation, intracellular signaling, and transcription factors expression. If JAK2 mutation, irrespective of the nature of the alteration, is known to be a crucial event for the disease to initiate, additional mutations seem to be markers of progression and poor prognosis. These discoveries have helped to characterize the complex genomic landscape of PV, resulting in potentially new adapted therapeutic strategies for patients concerning all the genetic interferences.

## 1. Introduction

Polycythemia vera (PV) is a chronic hematological disorder that involves the uncontrolled proliferation of myeloid blood cells within the bone marrow, also known as myeloproliferation, essentially affecting the erythroblastic lineage. First reported in 1892 [1], cases of PV were later classified and more precisely defined by the Polycythemia Vera Study Group (PVSG) in 1967 [2], then by the World Health Organization successively in 2001, 2008, and more recently in 2016. PV is one of the myeloproliferative neoplasms (MPNs) that also includes other entities such as essential thrombocythemia, primary myelofibrosis, and chronic neutrophil leukemia. Like all MPNs, PV is characterized by uncontrolled cell proliferation. PV is a myeloproliferative disorder characterized by erythroid hyperplasia, but also myeloid leukocytosis, thrombocytosis, and splenomegaly [3]. In very rare cases, PV and other MPN have affected multiple members of the same family, suggesting that genetic factors may play a role in the development of the disorder in addition to acquired gene mutations. Indeed, familial cases of PV appear in a subset of cases that present additional inherited genetic factors [4]. The main cause associated with PV development is a malignant genetic change within a single cell of the bone marrow, which is why PV is described as a clonal disorder. However, the underlying reason that this acquired malignant change occurs remains unknown. It has been established that 98 percent of patients with PV have a driver mutation of the JAK2 gene [5]. The remaining patients mainly present alterations in the pseudokinase domain of the JAK2 gene in the 12th to 15th exons [6] listed in Table 1. Within a small percentage of individuals with PV, additional “non-phenotypic driver mutations” can occur either in the TET2, DNMT3a, or ASXL1 genes [7]. More recently, other genetic abnormalities have been highlighted as potential sources of PV development, such as modifications of the NF-E2 [8] and LNK (SH2B3) genes [9]. 

Next-generation sequencing (NGS) arises as a major tool in defining the mutational landscape of many pathologies and appeared as helpful to refine disease diagnosis. Hematological malignancies, and especially chronic myeloproliferative neoplasms (MPNs) such as PV, takes part in it because of the point mutations detection ability of these techniques (e.g., JAK2 V617F). Over the past, it has been reported that JAK2 V617F allelic burdens/JAK2 exon 12 mutations were assessed by several molecular techniques such as conventional Sanger sequencing, pyrosequencing, or allelic discrimination [10,11,12,13,14]. However, greater specificity/sensitivity has been observed with more common quantitative polymerase chain reaction (qPCR) [10,11,12,13,14,15] and/or droplet digital PCR (ddPCR) [15,16]. Furthermore, myeloid neoplasm-associated mutations (MNAMs) [17]/additional (non-driver) mutations were found in PV patients thanks to Targeted Capture Sequencing (TCS) technology [5,18,19,20,21,22,23]. Since then, TCS has appeared as a template for this application [23]. All these mutations are listed below in the dedicated portion and summation in Table 2.

## 2. Mutational Landscape: Hallmarks of PV Genetic Background

### 2.1. Driver Mutations: JAK2

According to studies, researchers have determined that most individuals with polycythemia vera have a variation in the JAK2 gene [5,7,24,25,26]. 

The JAK2 gene encodes for the Janus kinase 2 protein, which is known to be a very powerful driver of cell growth. This protein is constitutively associated with the erythropoietin receptor and allows the phosphorylation of this receptor in physiological conditions, thereby enhancing the activation of proliferating molecular signalization [26]. In people with polycythemia vera, the JAK2 gene is overactive because of the underlying genetic change. A change in the DNA of a single hematopoietic stem cell causes the abnormal cell to reproduce continually, eventually becoming the predominant hematopoietic stem cell in the bone marrow. Because the JAK2 gene is overactive, it leads to enhanced activity of STAT signaling, notably through the STAT5 and STAT3 axes, which are crucial players in cell proliferation [27]. These mutated cells, derived from the original defective hematopoietic stem cell, continue to grow and divide even in the absence of erythropoietin. Erythropoietin secretion is thus downregulated. Multiple genetic alterations have been reported during the research. Most of these affected the pseudokinase domain of the JAK2 gene within exons 12 to 15, as described below and summarized in Figure 1 and Table 1.

#### 2.1.1. Canonical JAK2 Mutation: JAK2 V617F/Exon 14 Mutations

The most frequent variation in the JAK2 gene is the gain of function mutation val617-to-phe (V617F; 147796.0001), which was first reported by William Vainchenker’s team (Institut Gustave Roussy, Villejuif, France) in 2005 [26], and then confirmed by other research teams [24,25,28]. This unique amino acid substitution on the 14th exon leads to constitutive phosphorylation activity that promotes cytokine hypersensitivity and growth factor independence, as well as conferring a proliferative advantage to the mutated clones. Later, it was suggested that the principle of homozygous or heterozygous V617F mutation is associated with a distinct clinical phenotype and outcome [54]. Thus, JAK2 V617F homozygote PV patients (around 30% depending on the cohorts) displayed a significantly higher hemoglobin level, increased incidence of pruritus, stimulated erythropoiesis and myelopoiesis, higher prevalence of splenomegaly, and increased progenitor cells in peripheral blood associated with a higher risk of fibrotic transformation in comparison with their heterozygote counterparts [5,29,30]. 

JAK2 gene exon 14 is not exempt from the presence of other mutations. Among them, the complete absence of the 14th exon (Δ exon 14) is most described in the literature. This variant leads to a truncated JAK2 protein and is most likely due to exon skipping after alternative splicing. It should be noted that Δ exon 14 JAK2 mutant is more frequent in V617F-negative patients, in whom it might contribute to leukemogenesis [31].

Missense alterations have also been observed within exon 14 in PV patients, including V617I, C618R [31], L611V [32], and rarely C618F [35], H606Q, and H608Y mutations [31]. Moreover, L611S mutation, previously described alone in thrombocytosis and lymphoblastic leukemia [55,56], was recently identified in association with V617F mutation in a PV case report [33]. As different JAK2 mutants may have different JAK2 activity, the JAK2 mutational status may influence subclone outcome and affect the disease phenotype. More specifically, it is worthy of mention that a V617I alteration, like V617F, was shown to induce cytokine independence and constitutive downstream signaling [6,34], while L611V mutation trigger STAT signaling in PV patients [32].

It should also be noted that, while they are described in PV patients, most of the atypical JAK2 mutations within exon 14 have been associated with the profile of essential thrombocythemia. 

Although the great majority of PV patients present a V617F alteration within the JAK2 gene with an allele burden of 46% [54], genome sequencing among patient cohorts has highlighted other mutation profiles in other exons. 

#### 2.1.2. Non-Canonical JAK2 Mutations: JAK2 Exon 12, 13 & 15 Mutations

##### JAK2 Exon 12

JAK2 exon 12 mutations were first reported in PV JAK2V617F-negative patients in 2007 by genome analysis of peripheral blood cells in PV patients [39]. Despite the phenotypical difference, those alterations in the 12th exon are reported to mimic the outcome of JAK2 (V617F)-positive PV [57]. So far, up to 30 different mutations have been reported in JAK2 exon 12 mutated PV patients [36]. Despite the mutation, two-thirds of the patients had isolated erythrocytosis, and one-third had erythrocytosis plus leukocytosis and/or thrombocytosis [57]. Compared to JAK2 V617F-positive PV patients, those with exon 12 mutations had significantly higher hemoglobin levels, reduced serum erythropoietin levels, and lower platelet and leukocyte counts at diagnosis, but a similar incidence of thrombosis, myelofibrosis, leukemia, and death [38]. A recent study reported a younger age in JAK2 exon 12 PV patients and a similar prognosis to JAK2 V617F mutated PV patients [58]. Documented high-frequency JAK2 exon 12 mutations include in-frame deletions, missense, and tandem point mutations summarized as p.539L substitutions, p.E543 deletions, p.547 duplications, and the others, including the p.[N542-E543del] within 39.6% of patients, p.[F537-K539delinsL] or p.[H538QK539L] [37,40,41,42,43,44,45,46,47,48]. Whereas JAK2 V617F mutations are typically homozygous in patients with PV, JAK2 exon 12 mutations are often heterozygous. In addition, exon 12 mutations can induce cytokine-independent hypersensitive proliferation in erythropoietin-expressing cell lines and are sufficient to develop a PV-like phenotype in a murine model [59]. 

##### JAK2 Exon 13

Among the mutations identified as responsible for the development of PV, some have been highlighted in the 13th exon of the JAK2 gene. Several mutations in the pseudokinase domain coding region are described in the literature, specifically R564L, R564Q, V567A, G571S, G571R, L579F, H587N, S591L, and F557L (with frameshift and early termination) [31]. In particular, the G517S mutation, identified in case reports [51], is thought to alter the most important autophosphorylation site that contributes to the downregulation of JAK2 activity [52], leading to its constitutive activation without enhancing STAT5 signaling. The lack of strong molecular pathway activation due to that mutation suggests that JAK2 G517S is probably insufficient to trigger PV development and is unlikely to be the sole driver of abnormal erythropoiesis [51,53]. 

##### JAK2 Exon 15

The only JAK2 15th exon mutations reported in the literature are the I645V and L642P missense alterations [31]. 

Hence, there is significant interest in the contribution of JAK2-independent signaling in MPNs, particularly given that the same JAK2 mutation can lead to diverse disease phenotypes, and since JAK inhibitor therapy is limited in eradicating malignant clones. Thus, the acquisition of additional JAK2-independent events within the progenitor cell is presumably important for disease initiation and/or development.

### 2.2. MNAMs 

Since the emergence of high-throughput NGS analyses, many additional somatic mutations have been added to the main genetic aberrations previously reviewed. The affected genes highlight a new complex network in the mutational landscape of PV, and alterations are often found in a distinct clone from the one containing the initial driving mutation. Several molecular mechanisms, signaling pathways, and their downstream factors are impacted. As described below, it was found that these genes belong to alternative splicing (SRSF2, SF3B1, U2AF1, ZRSR2), epigenetic (TET2, DNMT3A, IDH1, IDH2, ASXL1, EZH2), miRNA deregulation, intracellular signaling (SH2B3, NF1, NRAS, KRAS, CBL, FLT3, PPM1D, ERBB) and several transcription factors (NF-E2, TP53, RUNX1, CUX1, ETV6) summarized in Table 2. Thanks to the constant upgrading of technology, these discoveries have helped to characterize the complex genomic landscape of PV.

#### 2.2.1. Mutations Involved in Alternative Splicing (SRSF2, SF3B1, U2AF1, ZRSR2)

Components of the pre-messenger RNA splicing machinery are frequently mutated in myeloid malignancies at various frequencies. Alterations in genes involved in mRNA maturation splicing, spliceosome components, a multimegadalton ribonucleoprotein complex involved in the maturation of gene encoding mRNAs [111], are not rare events in general MPNs [106,112,113]. 

Mutations have been reported in various genes including splicing factor 3B subunit 1 (SF3B1) [61], serine/arginine-rich splicing factor 2 (SRSF2) [22], U2 small nuclear RNA auxiliary factor 1 (U2AF1) [62] and zinc finger RNA binding motif and serine/arginine-rich 2 (ZRSR2) [19]. These mutations, which involve alternative splicing affections, add to the complexity of the mutational landscape of PV [19,60]. Interestingly, those additional mutations seem to enhance survival prediction in PV and can contribute to the identification of patients at risk for fibrotic progression [22]. 

#### 2.2.2. Mutations Involving Epigenetic

Among the increasing panel of additional mutations that might be implicated in the development or the progression of hematological malignancies, genetic alterations affecting epigenetic regulators, specifically DNA methylation and chromatin modification genes including TET2, DNMT3A, IDH1/2, ASXL1, and EZH2, were described as more frequent in PV than in other MPNs [114].

##### DNA Methylation (TET2, DNMT3A, IDH1, IDH2)


**TET2**


Over the last decade, investigation of TET2 (Ten-Eleven Translocation 2) gene function and mutation status have become of increasing interest in the field of hematology [63]. This heightened interest was sparked by the discovery that a TET2 mutation was associated with the development of hematological malignancies through its regulatory role in lineage commitment. Loss of TET2 function leads to dysregulated gene expression in hematopoietic stem cells and has been considered as a potential initiation step of myeloid and lymphoid malignant transformation in mice [64]. TET2 gene mutations include frameshift, generated stop codons, in-frame deletion, and amino acid substitutions of highly conserved residues [63]. In humans, analysis of the TET2 gene in bone marrow cells from 320 patients with myeloid cancers identified TET2 defects in 13 patients with polycythemia vera, all of whom also displayed the JAK2 V617F mutation [65]. Moreover, it has been reported that TET2 mutations are probably associated with more than 20% of PV cases [66].

Furthermore, the prognosis of TET2 mutation in hematologic malignancies has been controversial, and the detailed mechanism of TET2 in the promotion of malignancy needs to be further explored [63].


**DNMT3A**


DNMT3A (DNA methyltransferase 3A) is a de novo DNA methyltransferase that has recently gained relevance because of its frequent mutation in a large variety of hematologic malignancies. Because of its regulating role in somatic stem cell differentiation, the loss of DNMT3A activity leads to the self-renewal of cells rather than their differentiation or maturation [115]. 

Somatic DNMT3A mutations in the terminal exon were initially reported at low frequency in PV (2.7%) [67], but another study reported a higher prevalence of around 9% in their cohort, including 3 somatic DNMT3A mutations. One already known mutation, called R882, was observed, and the other two were novel frame-shift mutations at codon K456 [68]. Like for ASXL1, researchers assume that DNMT3A somatic mutations could appear prior to key driver mutations such as JAK2V617F and therefore be crucial players in at least the initiation phase of the disease.


**IDH1/IDH2**


The IDH gene encodes enzymes that catalyze oxidative decarboxylation of isocitrate to α-ketoglutarate, contributing to cellular protection from oxidative stress. Mutant IDH turns isocitrate affinity toward α-ketoglutarate, intracellular changes that facilitate oncogenic pathways [116]. It is worth noting that the most described mutations are heterozygous and occur mostly as point missense mutations at residues R132 in IDH1 and R140 or R172 in IDH2 [69].

IDH1 and IDH2 mutations were reported in a multi-center study involving 427 PV-patients, with a proportion of 2% of cases regardless of the presence of JAK2 and with no adverse effect on survival [70]. The most frequently described mutation within PV patients seems to be the IDH2R140Q alteration [69,70]. Moreover, IDH1/2 gene mutations might be a relevant criterion to forecast AML transformation [117]. 

##### Histone Modifications (ASXL1, EZH2)

Other mutations described in polycythemia vera include ASXL1 and EZH2. The effect of the mutation order in MPN has been clearly shown to alter clinical features and clonal evolution [92,118]. When looking at the temporal sequence of somatic mutation acquisition, it seems that while ASXL1 or EZH2 mutation acquisition prior to JAK2 V617F leads to the development of ET or primary MF, the late emergence of those mutations was more likely to constitute a PV phenotype in patients [114]. It is interesting to note that both ASXL1 and EZH2 are key regulators of chromatin silencing agents and are known to play a role in MPN initiation and disease progression [119].


**ASXL1**


ASXL1 (Additional sex combs like transcriptional regulator 1) belongs to a gene family responsible for maintaining the activation and silencing of proteins that are critical regulators of developmental genes by controlling the chromatin structure [120]. ASXL1 mutations mainly occur in exon 12, leading to a prematurely truncated protein that lacks its PHD domain (C-terminal plant homeodomain), which is involved in the interaction between proteins, therefore compromising the formation of chromatin modifier complexes [121].

Disruption of ASXL1 through inactivating mutations is identified only rarely in cases of PV, i.e., in around 4–7% [71,72,73]. Nonetheless, four inactivating somatic mutations in ASXL1 were identified by sequencing the peripheral blood cells of JAK2 V617F-positive PV patients. There was a higher prevalence of 22% in the cohort, and there were two frameshift and two nonsense mutations [68]. All four loss-of-function mutations were identified in exon 12. This mutation rate was 6-fold higher than previously reported and is like other MPNs. The role of ASXL1 in hematopoiesis is poorly understood, and its role in MPN is still under investigation. However, it may play a role in the progression of polycythemia vera and essential thrombocythemia to PMF, considering that this mutation is rare in essential thrombocythemia and polycythemia vera (less than 7%) but more frequent in PMF (19–40%) [74]. ASXL1 variants have been significantly associated with inferior survival [5]. In addition, researchers hypothesize that ASXL1 mutations could be new driver genes alterations, and pre-JAK2 mutations still further describe signatures of clonal evolution during PV progression in some patients.


**EZH2**


Enhancer of zeste 2 polycomb repressive complex 2 subunit (EZH2) encodes the catalytic subunit of the polycomb repressive complex 2 (PRC2), a highly conserved histone methyltransferase that influences stem cell renewal by epigenetic repression of the genes involved in cell outcome [122]. EZH2 has oncogenic activity, and its overexpression has been causally linked to differentiation blocks in epithelial tumors. Notably, the mutations we identified resulted in premature chain termination or direct abrogation of histone methyltransferase activity, suggesting that EZH2 acts as a tumor suppressor for myeloid malignancies [123].

Mutations in EZH2 have been described in patients with various hematologic malignancies, including approximately 3% of PV [75]. A screening of 518 patients suffering from PMF and post-PV/TE myelofibrosis found the EZH2 mutation in nearly 1% of post-PV myelofibrosis without significantly impacting overall survival [76]. However, other studies suggest that EZH2 loss of function mutation may significantly worsen survival, especially when considering subjects with homozygous mutations compared to heterozygous patients [75].

##### miRNA Deregulation

Apart from genetic alterations, another aspect is of great interest in the understanding of the development and management of PV. MicroRNA (miR) are non-coding 18–22nt RNA that regulate gene expression either by destabilizing target mRNA or inhibiting protein translation [124]. Studies demonstrate that deregulated miR may be important in determining the PV phenotype as dysregulation occurs in PV CD34+ cells [82,125,126]. Notably miR-451 [83], miR-150 [126], miR-28 [78], miR-125b-5p and miR-125a-5p [79], miR-182 and miR-342 [77] expression have been identified as altered in PV patients. After this observation, researchers looked deeper to identify candidate genes involved in the miRNA regulation network that might be altered in PV patients. 

The analysis highlighted significantly up- and down-regulated genes after miRNA modulation, leading to the determination of potential targets that might be directly controlled by miRNAs. Among miR-451 candidates, iron homeostasis (FTH1) and hematopoiesis-related genes (RUNX1) were included, as were cMYB, BNIP3L, p27, and EPOR, major proteins involved in the regulation of erythroid maturation and cell cycle [127]. A correlation between high miR-28 and MPL down-modulation has also been observed in PV patients. miR-28 was expressed in 50% JAK2 V617F-positive PV patients, potentially acting as an inhibitor of MPL translation and other major proteins for megakaryocyte differentiation. It is suggested that the expression of miR-28 might play an important role in the pathogenicity of MPNs, either as part of negative feedback of myeloproliferation or as a regulator of disease phenotype [78]. 

Moreover, upregulated expression of miR-125 has been correlated with platelet counts and the cytokine hypersensitivity of bone marrow hematopoietic progenitors without a significant link with the JAK2 allele burden [79]. While miR-182 upregulation in PV granulocytes is associated with JAK2 V617F allele burden [82], progressive miR-150 and miR-342 downregulation during erythropoiesis have been inversely correlated with JAK2 V617F allele burden in PV patients [77,81]. Concerning miR-143, a significant difference in expression has been observed for PV-patients, associated with augmented platelet count and JAK2 V617F allele burden. Interestingly, miR-143 expression was higher in homozygous compared with heterozygous JAK2 V617F patients. It has been suggested that miR-143 up-regulation could lead to a decreased expression of KRAS, resulting in exaggerated erythropoiesis [80]. Additionally, miR-145, which promotes erythrocyte differentiation of the megakaryocyte-erythroid progenitor cell [128], is upregulated in CD34+ and the erythroid-lineage cells of PV patients [129]. 

As a result, these data tend to confirm the role of miRs as regulators of erythropoiesis. Suggesting that aberrant expression of miRNAs may contribute to abnormal erythropoiesis and favor the emergence of phenotype and an miR erythroid signature such as that of PV [83,129].

#### 2.2.3. Mutations Involved in Intracellular Signaling (LNK/SH2B3, NF1, NRAS/KRAS, CBL, FLT3, ERBB)

Intracellular signaling alteration, as an important hallmark of all cancer types, might not only contribute to the malignant transformation of cells but also promote disease progression and reduce survival rate. Indeed, mutations within key cellular actors such as LNK/SH2B3, NF1, NRAS/KRAS, CBL, FLT3, PPM1D, ERBB play a major role as additional factors triggering PV development and progression. 


**LNK/SH2B3**


LNK (Lymphocyte adaptor protein) is an important control switch in hematopoietic stem cells. The global inhibitory effect of LNK was first demonstrated on the proliferation of HSCs, lymphoid and myeloid progenitors, and mature cells, mainly from erythroid and megakaryocytic lineages [130,131,132]. LNK is a negative regulator of JAK2 in stem cells and contributes to the regulatory pathway controlling stem cell self-renewal and quiescence. In addition to its role in early hematopoiesis, LNK can also modulate signaling mediated by lineage-specific cytokines, including TPO and EPO, thus controlling megakaryocytic and erythroid development, respectively. Indeed, through its SH2 domain, LNK negatively modulates EPO receptor (EPOR) signaling by inhibiting three important pathways in primary erythroblasts: JAK2/STAT5, AKT, and MAPK [133].

Consistent with the negative regulatory role of LNK in hematopoiesis and the myeloproliferative phenotype of LNK-deficient mice, mutations in the SH2B3 gene have been reported in approximately 7% of PV patients [5,84]. Most of the SH2B3 mutations identified are missense mutations targeting all exons, resulting in a reduced activity level. SH2B3 mutations are thought to be implicated in the induction or/and the development of the disease phenotype. In addition, an SH2B3 mutation was observed in one JAK2V617F-negative patient with PV syndrome [85], but it most often coexists in patients with a JAK2V617F-mutated PV pathology [9], or other genes (TET2, ASXL1), indicating that they may also cooperate with other mutations to induce the associated phenotype. Some researchers even suspect that a double alteration of JAK2 and SH2B3 genes would synergize and further favor the transformed phenotype of mutant cells [86]. Indeed, JAK2V617F expression in LNK-deficient murine bone marrow cells rendered the cells significantly more sensitive to transformation by JAK2 oncogene compared to wild-type cells in clonogenic assays and accelerated the onset of JAK2V617F PV in mice [87,88].


**NF1**


The tumor suppressor gene Neurofibromin 1 (NF1) codes for a multifunctional protein, neurofibromin. Neurofibromin is known to impact a large panel of cellular processes, including proliferation, growth, division, survival, and migration due to its action in several cell signaling pathways (e.g., Ras/MAPK, Akt/mTOR, ROCK/LIMK/cofilin, and cAMP/PKA pathways) [134]. Because of its multifunctional implication in master cell pathways, mutations that affect this gene cause tumor predisposition syndrome neurofibromatosis type 1 and have been observed in multiple cancer types, notably in hematologic malignancies, including myelodysplastic syndromes [135].

In PV, NF1 emerges as a frequently mutated gene (around 15%) in multiple studies [68,89]. The most frequent acquired mutations within the NF1 gene resulted in a loss of function of the mature protein. Loss of NF1 causes Ras constitutive activation, stimulating induction-independent cell proliferation [90]. Additionally, an NF1 knockout mouse model has developed an MPN phenotype [136]. Among the two cases reported in the study, one patient had wild-type JAK2 protein, and the other presented homozygous V617F mutation. It is hypothesized that the loss of NF1 could be a late event likely to give additional growth potential even to cells homozygous for JAK2-V617F and might contribute to disease progression towards secondary myelofibrosis [73,89,91].


**NRAS/KRAS**


RAS proteins are small GTPases that act as molecular switches to transduce signals from activated receptors. When RAS proteins are in an active state, they bind to and activate a range of downstream effector proteins, resulting in diverse cellular outcomes like cell proliferation, survival, differentiation, and neoplastic transformation [137,138]. Mutations that result in constitutive activation of RAS proteins are associated with ~30% of all human cancers [139]. However, different RAS oncogenes are preferentially associated with different types of human cancer. In myeloid malignancies, NRAS mutations are more frequent than KRAS mutations [140]. Specifically, in MPNs, heterozygous missense mutations have been observed, especially in codons 12, 13, and 61, leading to a constitutive activation of growth signaling [60]. NRAS mutations have been found to have a prognostic role because patients harboring NRAS/KRAS alterations face a significantly reduced overall survival. Also, patients with NRAS/KRAS mutations had more non-driver mutations [141]. Those mutated forms have been observed in patients with PV or post-PV myelofibrosis [92,142], in 3.5% for NRAS and 1.3% for KRAS. Moreover, it is suggested that KRAS/NRAS mutations result from sub-clonal events acquired during the course of the disease [142]. 


**CBL**


CBL gene codes for a protein that acts as a negative regulator of signaling pathways and, notably, cell proliferation. In the past, mutated CBL was shown to functionally and genetically act as a tumor suppressor and has been observed in hematologic neoplasms [143,144]. These CBL alterations cause a loss of function of the mature protein resulting in dysregulation of downstream targets and increased cell proliferation rates [93]. During in vitro experiments, multiple groups have observed that hematopoietic stem/progenitor cells modified with the mutated form of the CBL gene (inactive) had augmented sensitivity to a broader spectrum of cytokines [144] and favored myeloproliferation, resulting in activated JAK/STAT and PI3K/AKT signaling in murine models [145]. These data actively suggest the pathogenic importance of these alterations and their role in inducing an oncogenic phenotype in various cell lines and their independence to growth factors. Recurrent change within exon 12 (S675C) has been observed in 1.5% of JAK2 V617F-positive PV patients [92,93]. It is to be noted that CBL mutations and JAK2 V617F seemed at first to be mutually exclusive [146]. However, a similar frequency of CBL mutations in both V617FJAK2-positive and V617FJAK2-negative patients has been observed [93]. 


**FLT3**


FMS-like tyrosine kinase 3 (FLT3) is a member of the class III receptor tyrosine kinase family, which plays an important role in developing multipotent hematopoietic stem cells and lymphoid cells [147,148]. FLT3 is expressed at high levels in numerous hematopoietic malignancies [94]. The most described abnormality on FLT3 is an internal tandem duplication (ITD) resulting from duplication of a portion of the gene. It has been established that retroviral transduction of FLT3-ITD into primary murine bone marrow cells results in a myeloproliferative phenotype in a bone marrow transplant assay [149]. It was then abrogated in a rescue assay, proving an absolute requirement for FLT3 kinase activity in developing myeloproliferative disease in this model. Altered FLT3 has been reported for one patient suffering from PV [94] and in post-PV myelofibrosis [68]. It has been suggested that the acquisition of this mutation is a rather late event in the course of the disease. 


**ErbB**


The ErbB receptor family, including the epidermal growth factor receptor (ErbB/EGFR), represents a group of receptor tyrosine kinases (RTKs) [150]. The activated ErbB receptors bind to many signaling proteins and stimulate the activation of many signaling pathways, including the Ras-Raf-Mek-ERK, PI3K-Akt-Tor, PLC-γ1, STAT, and Src pathways [151] with established roles in cancer, both contributing to tumorigenesis and the progression of the disease [152]. Somatic gain-of-function mutations affecting ErbB are observed in myeloproliferative neoplasms (MPN). Thus, mutant-related constitutive activation of this group of receptors, via multiple mechanisms, may contribute to clonal growth and survival of the JAK2V617F disease clone in MPN. An ErbB1/EGFR somatic mutation (C329R) has been identified in a patient with JAK2V617F-positive PV. This substitution leads to the formation of a ligand-independent covalent receptor dimer associated with increased transforming potential. Consistent with a role in clonal expansion and PV pathogenesis, the C329R mutation results in a loss of erythroid lineage markers and reduced EPO-induced differentiation both in an erythroid differentiation model and in a PV patient cell sample. Because of ErbB/EGFR crossed impact with the JAK2 pathway on cell signaling, this rare event likely cooperates with JAK2 signaling to influence the growth and lineage properties of the PV clone [95].

#### 2.2.4. Mutations Affecting Transcription Factors (NF-E2, PPM1D, TP53, RUNX1, CUX1, ETV6)

All components of cells are produced via DNA reading, itself regulated by multiple transcription factors, so it is reasonable to surmise that genetic abnormalities affecting these transcription factors might have a leading role in disease development, notably in hematologic malignancies such as PV. Here we will review the impact of NF-E2, PPM1D, TP53, RUNX1, CUX1, ETV6 alterations within the course of PV. 


**NF-E2**


Researchers have identified different somatic insertion or deletion mutations in the NF-E2 (Nuclear Factor Erythroid 2) gene in three patients V617F-positive PV (2.1% of PV cases) [8]. Other studies highlighted that 8.7% [96] and 4.5% [17] of PV patients present NF-E2 alterations. For one of those studies, the authors demonstrated that NF-E2 mutations were heterozygous with a variant allele frequency (VAF) of 34.5% ± 14.7 in PV [96]. The NF-E2 transcription factor is found almost exclusively in hematopoietic progenitors and cells of the erythroid/mega/mast cell trilineage. NF-E2 is involved in regulating globin gene transcription, and in addition, is essential for normal platelet production [153]. In vitro studies showed that the truncated NF-E2 mutant proteins were unable to bind DNA and had lost reporter gene activity, hence enhancing the gene activity of its targets. Hematopoietic cell colonies grown from 3 patients showed that the NF-E2 mutation was acquired after the JAK2 mutation. Further cellular studies indicated that an NF-E2 mutation conferred a proliferative advantage of cells compared to cells carrying only the JAK2 mutation [8]. Cells carrying mutant NFE2 displayed an increase in the proportion of cells in the S phase, consistent with enhanced cell division and proliferation, and this was associated with higher levels of cell cycle regulators. These findings were replicated in mice carrying NF-E2 mutations, who developed thrombocytosis, erythrocytosis, and neutrophilia [154]. Analysis of samples from PV patients showed a 2 to 40-fold overexpression of NF-E2 transcription factor. In bone marrow, an increment of the latter is observed in megakaryocytes, erythroid, and granulocytic precursors with the development of erythropoietin-independent erythroid colonies and probably control of the lineage commitment. Finally, the level of NF-E2 overexpression may contribute to determining both the severity of erythrocytosis and the presence of thrombocytosis [97]. 


**PPM1D**


The phosphatase, Mg^2+^/Mn^2+^-dependent 1D (PPM1D) gene encodes a protein that regulates the DNA damage response pathway by inhibiting tumor-suppressors actors [155]. The most commonly described mutations rely on truncating alterations within the exon 6 of the PPM1D gene [98], which is thought to confer a proliferative advantage to mutant clones. These alterations are often related to a “treatment-response” mutation in studies focused on hematologic malignancies [155]. Truncating mutations in the terminal exon of PPM1D have been identified in 1.9% of an MPN cohort, and PPM1D was thus the eighth-most mutated gene in myeloproliferative neoplasms. The mutated form of PPM1D was subclonal to JAK2V617F in a patient with polycythemia vera [17]. It is to be noted that the order in which mutations are acquired in myeloproliferative neoplasms has previously been shown to influence disease phenotype. In this context, PPM1D mutations have been shown to appear significantly later in the course of the disease. 


**TP53**


One cellular pathway of particular interest in MPN disease progression involves the tumor suppressor protein p53 (TP53). This gene encodes a DNA-binding protein that responds to DNA damage by either inducing cell cycle arrest or apoptosis. Thus, the loss or inactivation of TP53 plays a critical role in the pathogenesis of many cancers [156]. While TP53 alterations might be observed in response to treatment or with an intrinsic link with age [17], multiple teams have demonstrated that chromosomal abnormalities affecting the TP53 gene (exon 5 to 8), notably missense mutations or duplication, did not include any association between a specific TP53 mutation type or any MPN subtype [157]. However, studies have found TP53 mutations in 70% of patients with PV-related MF [99] and approximately 8% of PV patients [158] with a very low allele burden [100], and TP53-mutated subclones within JAK2 or CALR-mutated populations [158]. The acquisition of TP53, acquired significantly later in the disease [17], appears to be a particularly unfavorable event, and loss of heterozygosity was invariably associated with disease progression [92,101]. Interestingly, in vivo experiments in a Jak2 V617F-positive mice model have shown that p53 loss is sufficient for inducing leukemic transformation after the PV phase [159]. 


**RUNX1**


The Runt-related transcription factor 1 (RUNX1) is a transcription factor essential for normal hematopoiesis, acting as a key regulator of hematopoietic-specific genes and contributing to the transformation of hematopoietic progenitors by altering the DNA binding potential of this critical transcription factor in hematopoiesis [160]. RUNX1 is over-expressed in granulocytes and primary erythroid progenitors of chronic phase MPN patients [161]. It is one of several genes in which mutations (missense, frameshift, and nonsense) have been identified in blasts from MPN patients who have progressed to leukemia, suggesting that RUNX1 has a role in MPN disease progression [102,103]. Moreover, short-term transformations were mostly characterized by mutations in RUNX1 [100]. These mutations may be acquired only at the time of leukemic transformation or are already detectable during the chronic phase, although at a very low level [92], and they are associated with an adverse prognosis [19]. Altered RUNX1 gene transcripts were increased in the BFU-Es and granulocytes of PV patients, accompanied by augmented HIF signaling, suggesting the cytokine-hypersensitivity of erythroid progenitors [104]. While RUNX1 alterations are infrequent in PV (2%) [5] and found in 10–37% of post-MPN AML patients [102,103,162,163,164], they were associated with an adverse mutational effect, confirming the pathogenic relevance for leukemic transformation in MPN [19]. 


**CUX1**


The CUT-like homeobox 1 (CUX1) is described as a tumor-suppressor gene [165] that is thought to act on DNA damage repair in response to oxidative stress. Moreover, copy number and expression of the CUX1 gene increase in many cancers and are associated with poorer prognosis. Some findings implicate the DNA repair dysfunction resulting from CUX1 alterations in the pathogenesis of MNs [166]. On the other hand, it was demonstrated in a mice model that CUX1 knockdown promotes PI3K signaling, driving the exit of HSC from quiescence and proliferation, and resulting in exhaustion [167]. RUNX1 has been identified in a JAK2 V617F-positive PV patient [92] and post-PV myelofibrosis patients, and this alteration appears to be a sign of disease evolution and cell transformation [100]. 


**ETV6**


The translocation-Ets-leukemia virus 6 protein plays a crucial role in embryonic development and hematopoietic regulation [168]. ETV6 is also a versatile element at the center of a network of genes involved in hematologic malignancies through diverse molecular mechanisms, such as fusion with other genes and deletions [68]. ETV6 mutations (Missense/indel) in MPNs are associated with disease progression to AML (in <3% cases) [105]. 

#### 2.2.5. Other: CALR, MPLW515

While JAK2 mutations trigger most PV diseases in patients, a few cases of mutated forms of CALR and MPL have been reported in the literature. 


**CALR**


CALR encodes for calreticulin, a 46-kDa chaperone protein located in the lumen of the endoplasmic reticulum (ER). CALR has a key role in maintaining calcium homeostasis and protein folding [169]. Two CALR mutations have been reported: type-1 is a 52-bp deletion, and type-2 is a 5-bp insertion, resulting in mutant proteins that lose the ER-retention motif (KDEL) at the C-terminus [107] that is supposed to promote the extracellular JAK signaling pathway. Because 96% to 99% of PV patients harbor a JAK2 mutation, it seemed logical to assume that CALR mutations would be rare or absent [108,109,170,171]. CALR mutants were mostly type-1 and were found in peripheral granulocytes and BFU-E at diagnosis [108]. It is worth noting that CALR alterations occurred more commonly early in the disease [118]. 


**MPL**


Myeloproliferative leukemia virus oncogene (MPL) is known to encode the TPO receptor. The most commonly identified MPL acquired mutations are W515L (tryptophan-to-leucine substitution) and W515K (tryptophan-to-lysine substitution), which induce constitutive, cytokine-independent activation of the JAK-STAT pathway [110]. MPL mutations occurred more commonly early in disease [118]. MPL mutations have been observed in a few cases of JAK2 V617F positive PV and post-PV myelofibrosis [172], supporting previous data reporting that MPL mutations rarely occur in PV [5].

### 2.3. Genetic Associations: Pattern and Consequences

To go further in the understanding of this complex genetic landscape, some associations have been highlighted in NGS studies. Here, we summarized 7 PV patients cohorts in a graphical overview (Figure 2) [5,17,92,151,173,174,175] in which each patient’s mutational landscape and their own interplay between pathways/mutations are reported, as previously described. First, it is to be noted that, especially in the context of PV, the so-called driver mutations (JAK2, CALR, and MPL) are reported as mutually exclusive but also that approximately one-third of patients have MNAMs enhancing hence clonal heterogeneity [173]. Nonetheless, multiple mutations within driver genes can be observed in rare cases and are more frequently encountered in patients with low JAK2V617F allele burden [16], whether double mutated (JAK2/CALR or JAK2V617F/JAK2 Exon 12 or JAK2/MPL) [176] or even rarest triple mutated patients (JAK2V617F, CALR del52, and MPLS505N) [174]. Armed with the data collected in Figure 2, we summarized in Figure 3 and Figure 4 the MNAMs/pathways to which JAK2 mutations in PV are mostly associated. So far, we noted that JAK2 alterations mostly correlate with mutations within the epigenetic pathway and particularly with TET2 mutations, which are found in 37% of reported cases. Among the most frequent sequence variants/mutations, in PV, JAK2 is co-mutated with ASXL1 in 13%, with DNMT3A in 9%, NF-E2 in 7%, SH2B3 in 5%, and EZH2 in 4% of cases. Moreover, associations are observed between TET2/ASXL1 (28%), TET2/SH2B3 (46%), TET2/DNMT3A (17%), TET2/EZH2 (40%), TET2/NF-E2 (11%), ZRSF2/SH2B3 (57%) [5,17,92,175]. As described previously, as the type of mutation seems to have incidence on the disease course, the presence of 3 or more altered genes, independent of their nature, is a marker of bad prognosis for patients [177].

## 3. Conclusions

Since 2005, when the JAK2V617F driver mutation was identified in PV, many other associated genomic markers that play a role in the expression and progression of the disease have been observed. Transformation to acute leukemia or secondary myelofibrosis is usually associated with the acquisition of other molecular markers that generally confer a poor prognosis. The current routine use of new sequencing techniques makes it possible to better assess the genetic background of mutations associated with MPN by identifying markers of poor prognosis, resulting in an adapted long-term therapeutic strategy.

## Figures and Tables

**Figure 1 genes-13-00637-f001:**
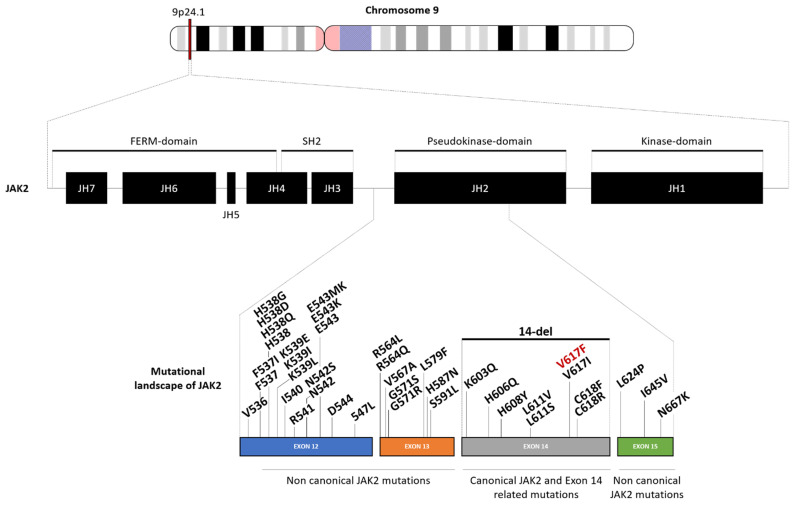
Schematic view of JAK2 gene structure and its mutational landscape in PV. JAK2 gene is found on chromosome 9 at 9p24.1 (4,984,390–5,129,948) location. It is composed of 7 homology domains (JH1 to JH7) that correlate to 4 functional domains: FERM, SH2, Pseudokinase, and Kinase domains. In PV, the mutations affected the region of JAK2 span from exon 12 to exon 15 and mainly belong to the JH2/Pseudokinase-domain. Abbreviation: del = deletion; FERM = four-point-one, ezrin, radixin, moesin; JH = Jak homology region; SH2 = src homology 2 domain.

**Figure 2 genes-13-00637-f002:**
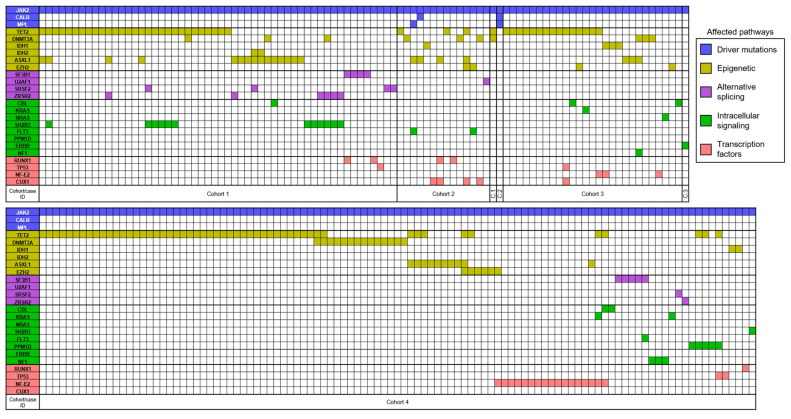
Graphical sum-up of 7 PV patients cohort/case reports (total = 205 patients). Each column represents one patient with all his mutational landscape divided by affected pathway (Driver mutations, Epigenetic, Alternative splicing, Intracellular signaling and Transcription factors). Cohort/case report references: Cohort 1 [5], Cohort 2 [176], Case report 1 [173], Case report 2 [175], Cohort 3 [92], Case report 3 [151] and Cohort 4 [17]. Abbreviations: C.1 = Case report 1; C.2 = Case report 2; C.3 = Case report 3.

**Figure 3 genes-13-00637-f003:**
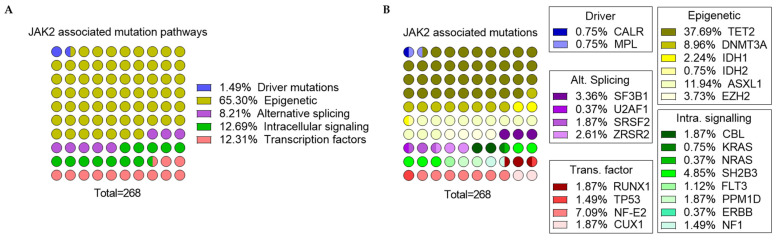
Overview of JAK2/MNAMs association in PV (total = 268 combinations) (based on Figure 2 related data). A visual dot plot representation of JAK2 associated mutations grouped by pathway (**A**) and individual affected gene (**B**) (each circle represents 1% of the total). Abbreviation: Alt. Splicing = Alternative Splicing; Intra. signaling = Intracellular signaling; Trans. factor = Transcription factor.

**Figure 4 genes-13-00637-f004:**
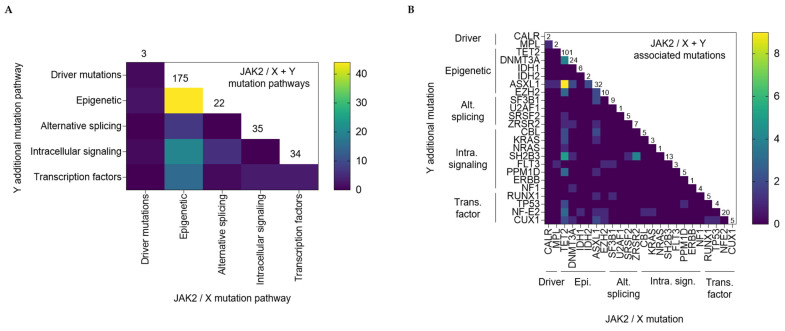
Heat map view of JAK2 paring with additional MNAMs in 7 PV patients cohort reports (Figure 2). (**A**) Heat map representation of JAK2/affected pathway association with another MNAM that belongs to one of the 5 affected pathways (e.g., the triple association of JAK2 + one of epigenetic affected gene + one of intracellular signaling affected gene is found in 20 cases among the total of 175 cases affected by a JAK2/Epigenetic association). The numbers above each colon represent the total cases of each JAK2/affected pathway (e.g., association of JAK2 + one of epigenetic affected gene is found in 175 cases). (**B**) Heat map representation of JAK2/affected gene with another MNAM (e.g., the triple association of JAK2 + TET2 + ASXL1 is found in 9 cases among the 101 cases affected by a JAK2/TET2 association.). The numbers above each colon represent the total cases of each JAK2/MNAM association (e.g., association of JAK2 + TET2 is found in 101 cases). Abbreviation: Alt. splicing = Alternative splicing; Epi. = Epigenetic; Intra. sign. = Intracellular signaling; Trans. factor = Transcription factor.

**Table 1 genes-13-00637-t001:** Summary of canonical and atypical JAK2 mutations in PV. NB: 96% to 99% of patients that suffer from PV are JAK2 mutated.

Canonical/Non Canonical	Location	Mutation	Comments		References
**Canonical JAK2** and **Exon 14** related mutations	Exon 14	*V617F*	Homozygous (around 30%) → higher PV related disorders and fibrotic transformation risk than heterozygote counterparts.		[5,24,25,26,28,29,30]
		Complete absence	Most likely due to exon skipping after alternative splicing. More frequent in *V617F*-neg. PV patients.		[31]
		*H606Q*	-		[31]
		*H608Y*			
		*L611V*	STAT signaling triggering		[32]
		*L611S*	Associated with *V617F* mutations.		[33]
		*V617I*	As V617F, induce cytokine independence & constitutive JAK2 downstream signaling.		[6,31,34]
		*C618F*	-		[35]
		*C618R*			[31]
**Non canonical/atypical** JAK2 mutations	Exon 12	V536-I546 dup11	Mut. freq.: 1.1%	First reported in JAK2*V617F*-neg. PV. Mimic the outcome of JAK2*V617F*-pos. PV patients. Often heterozygous.	[36]
		V536-F547 dup	Mut. freq.: 1.1%		[37]
		F537-I546dup10F547L	Mut. freq.: 1.1%		[36]
		F537IK539I	Mut. freq.: 1.1%		[38]
		F537-K539delinsL	Mut. freq.: 9.9%		[36,37,39,40]
		H538QK539L	Mut. freq.: 4.4%		[37,39]
		H538-K539delinsL	Mut. freq.: 3.3%		[36,41,42]
		H538-K539del	Mut. freq.: 1.1%		[37]
		H538DK539LI540S	Mut. freq.: 1.1%		
		H538G	Mut. freq.: 1.1%		[38]
		K539L	Mut. freq.: 7.7%		[37,38,39,41]
		K539E	Mut. freq.: 1.1%		[38]
		I540-E543delinsMK	Mut. freq.: 3.3%		[36,43]
		I540-E542delinsS	Mut. freq.: 1.1%		[44]
		R541-E543delinsK	Mut. freq.: 9.9%		[36,42,43,44,45]
		N542-E543del	Mut. freq.: 39.6%		[36,37,39,40,41,42,44,46,47,48,49,50]
		E543-D544del	Mut. freq.: 8.8%		[36,37,48]
		D544-L545del	Mut. freq.: 8.8%		[44]
		547insLI540-F547dup8	Mut. freq.: 1.1%		
**Non canonical/atypical** JAK2 mutations	Exon 13	*F557L*	With frameshift and early termination. *G571S*: • Alter the most important autophosphorylation site → downregulation of JAK2 activity. • Probably not sufficient to trigger PV development.		[31]
		*R564Q*			
		*R564L*			
		*V567A*			
		*G571S*			[31,51,52,53]
		*G571R*			[31]
		*L579F*			
		*H587N*			
		*S591L*			
	Exon 15	*L642P*	-		
		*I645V*			

Abbreviations: del = deletion; dup = duplication; ins = insertion; Mut. Freq. = Mutation frequency; neg. = negative; pos. = positive; PV = Polycythemia Vera.

**Table 2 genes-13-00637-t002:** Summary of additional mutations in PV.

Pathway		Affected Gene			Location	Comments	Frequency in PV	References
		Gene Symbol	Full Name	Alias				
**Alternative splicing**		SRSF2	Serine and arginine Rich Splicing Factor 2	SC35, PR264, SC-35, SFRS2, SFRS2A, SRp30b	17q25.1	Additional mutations that seem to enhance survival prediction in PV and can contribute to identifying patients at risk for fibrotic progression.	<3%	[3,60]
		SF3B1	Splicing Factor 3b subunit 1	MDS, PRP10, Hsh155, PRPF10, SAP155, SF3b155	2q33.1		5%	[3,60,61]
		U2AF1	U2 small nuclear RNA Auxiliary Factor 1	RN, FP793, U2AF35, U2AFBP, RNU2AF1	21q22.3		1–2%	[3,60,62]
		ZRSR2	Zinc finger CCCH-type, RNA binding motif and serine/arginine rich 2	URP, ZC3H22, U2AF1L2, U2AF1RS2, U2AF1-RS2	Xp22.2		1–2%	[3,19,60]
**Epigenetic**	**DNA methylation**	TET2	TET methylcytosine dioxygenase 2	MDS, IMD75, KIAA1546	4q24	Frame shift, generated stop codons, in-frame deletion, and amino acid substitutions of highly conserved residues.	>20%	[63,64,65,66]
		DNMT3A	DNA MethylTransferase 3α	TBRS, HESJAS, DNMT3A2, M.HsaIIIA	2p23.3	Terminal exon 3 somatic mutations	2.7% 9%	[67,68]
		IDH1/IDH2	Isocitrate DeHydrogenase (NADP(+)) 1/2	IDH1: IDH, IDP, IDCD, IDPC, PICD, HEL-216, HEL-S-26 IDH2: IDH, IDP, IDHM, IDPM, ICD-M, IDH-2, D2HGA2, mNADP-IDH	IDH1: 2q34 IDH2: 15q26.1	IDH1: R132 IDH2: R140 or R172	2%	[69,70]
	**Histone modifications**	ASXL1	ASXL transcriptional regulator 1	MDS, BOPS	20q11.21	4 inactivating somatic mutations in JAK2*V617F*-pos. PV patients (exon 12): 2 frameshift 2 nonsenses	4–7%	[68,71,72,73,74]
		EZH2	Enhancer of Zeste 2 polycomb repressive complex 2 subunit	WVS, ENX1, KMT6, WVS2, ENX-1, EZH2b, KMT6A	7q36.1	Resulted in premature chain termination or direct abrogation of histone methyltransferase activity.	3% (PV) 1% (post-PV MF)	[75,76]
**Epigenetic**	**miRNA deregulation**	let-7a	microRNA let-7a-1	LET7A1, let-7a-1, MIRNLET7A1	9q22.32	Down-regulation in granulocytes of PV patients. Correlations between aberrant expression of let-7a and JAK2*V617F* Mut. freq.	-	[77]
		miR-26b	microRNA 26b	MIRN26B, miR-26b, hsa-mir-26b	2q35	Up-regulation in platelets of PV patients.	-	
		miR-27b	microRNA 27b	MIR-27b, MIRN27B, miRNA27B	9q22.32	Up-regulation in platelets of PV patients.	-	
		miR-28	microRNA 28	MIRN28, miR-28, hsa-mir-28	3q28	Correlation between high miR-28 and MPL down-modulation → act as an inhibitor of MPL translation. Overexpression of miR-28 platelets (a fraction of PV & ET patients, wild type for JAK2).	50% (JAK2*V617F*-pos. PV)	[78]
		miR-30b miR-30c	microRNA 30b microRNA 30c	miR-30b: MIRN30B, mir-30b miR-30c: MIRN30C1, mir-30c-1	miR-30b 8q24.22 miR-30c 1p34.2	Down-regulation in reticulocytes of PV patients. Correlations between aberrant expression of miR-30c and JAK2*V617F* Mut. freq. (Inversely correlated with JAK2*V617F* allele burden).	-	[77]
		miR-125a-5p miR-125b-5p	microRNA 125a-5p microRNA 125b-5p	miR-125a-5p: hsa-miR-125a-5p, miR-125, hsa-miR-125a, miR-125a, MIR125A, miR-125a-5p miR-125b-5p: MIR125B1, miR-125b, miR-125b-5p, hsa-miR-125b-5p, MIR125B2, hsa-miR-125b, miR-125	miR-125a-5p: 19q13.41 miR-125b-5p: 11q24.1	Significant correlation between miR-125a-5p and platelet counts in PV patients.	-	[79]
**Epigenetic**	**miRNA deregulation**	miR-143	microRNA 143	MIRN143, mir-143	5q32	Up-regulation in mononuclear cells of PV patients. Correlations between aberrant expression of miR-143 and JAK2*V617F* Mut. freq. (Reflect JAK2*V617F* allele burden). Up-regulation → KRAS decreased expression → exaggerated erythropoiesis.	-	[77,80]
		miR-145	microRNA 145	MIRN145, miR-145, miRNA145	5q32	Up-regulation in mononuclear cells of PV patients.	-	[77]
		miR-150	microRNA 150	MIRN150, mir-150, miRNA150	19q13.33	miR-150 progressive downregulation (erythropoiesis) → inversely correlated with JAK2*V617F* allele burden.	-	[77,81]
		miR-182	microRNA 182	MIRN182, mir-182, miRNA182	7q32.2	miR-182 upregulation in PV granulocytes is associated with JAK2*V617F* allele burden.	-	[77,81,82]
		miR-223	microRNA 223	MIRN223, mir-223, miRNA223	Xq12	Up-regulation in mononuclear cells of PV patients.	-	[77]
		miR-342	microRNA 342	MIRN342, hsa-mir-342	14q32.2	miR-342 progressive downregulation (erythropoiesis) → inversely correlated with JAK2V617F allele burden.	-	[77,81]
		miR-451	microRNA 451	MIR451, MIRN451, mir-451a, hsa-mir-451, hsa-mir-451a	17q11.2	Up-regulation in mononuclear cells of PV patients.	-	[77,83]
**Intracellular signaling**		LNK/SH2B3	SH2B adaptor protein 3	IDDM20	12q24.12	Missense mutations targeting all exons, resulting in a reduced level of activity. Coexist in patients with JAK2*V617F* (one case in JAK2*V617F*-neg. PV patient)	7%	[5,9,84,85,86,87,88]
**Intracellular signaling**		NF1	NeuroFibromin 1	WSS, NFNS, VRNF	17q11.2	Loss of function of the mature protein → Ras constitutive activation 2 case reports: WT JAK2 protein/homozygous *V617F* mutation	15%	[68,73,89,90,91]
		CBL	Cbl proto-oncogene	CBL2, NSLL, C-CBL, RNF55, FRA11B	11q23.3	Recurrent change within the exon 12 (*S675C*) in JAK2 *V617F*-pos. PV patients. Similar frequency of CBL mutations in both JAK2*V617F*-pos. & JAK2*V617F*-neg. PV patients.	1.5%	[92,93]
		FLT3	Fms related receptor tyrosine kinase 3	FLK2, STK1, CD135, FLK-2	13q12.2	Internal tandem duplication (ITD) (most described). Only reported for a patient suffering from PV, but also in post-PV MF.	-	[68,94]
		ERBB	Epidermal Growth Factor Receptor	EGFR, ERBB1, ERRP, HER1, NISBD2, PIG61, mENA	7p11.2	ERBB1/EGFR somatic mutation (*C329R*) in JAK2*V617F*-pos. patient.	-	[95]
**Transcription factors**		NF-E2	Nuclear Factor, Erythroid 2	NFE2, p45	12q13.13	Somatic insertion or deletion mutations in 3 patients JAK2*V617F*-pos. Patients (after the JAK2 mutation) → proliferative advantages. 2 to 40-fold overexpression of NF-E2 in PV patients. Heterozygous	2–9%	[8,17,60,96,97]
		PPM1D	Protein Phosphatase, Mg^2+^/Mn^2+^ dependent 1D	IDDGIP, JDVS, PP2C-DELTA, WIP1	17q23.2	Most described mutations in exon 6 → proliferative advantages. Mutated form of PPM1D was subclonal to JAK2*V617F* in PV patients (appear significantly later).	2%	[17,60,98]
**Transcription factors**		TP53	Tumor Protein p53	BCC7, BMFS5, LFS1, P53, TRP53	17p13.1	TP53 mutations in 70% of patients with PV-related MF & 8% with PV. Very low allele burden/appear significantly later/loss of heterozygosity → disease progression. TP53-mutated subclones within JAK2 or CALR-mutated populations.	8% (PV) 70% (post-PV MF)	[17,92,99,100,101]
		RUNX1	RUNX (Runt-related) family transcription factor 1	AML1, AML1-EVI-1, AMLCR1, CBF2alpha, CBFA2, EVI-1, PEBP2aB, PEBP2alpha	21q22.12	Over-expressed in erythroid progenitors. Missense, frameshift, and nonsense in a blast from MNP. RUNX1 alterated gene transcripts ↗ in BFU-Es and granulocytes of PV patients + ↗ HIF. Rare in PV	2% (PV)	[5,102,103,104]
		CUX1	Cut like homeobox 1	CASP, CDP, CDP/Cut, CDP1, COY1, CUTL1, CUX, Clox, Cux/CDP, GDDI, GOLIM6, Nbla10317, p100, p110, p200, p75	7q22.1	In a case of JAK2*V617F*-pos. PV patients. Sign of disease evolution and cell transformation (post-PV MF).	-	[92,100]
		ETV6	ETS Variant transcription factor 6	TEL, TEL/ABL, THC5	12p13.2	Versatile element. In MPN → disease progression to AML (in <3% cases).	-	[105]
**Other**		CALR	Calreticulin	CRT, HEL-S-99n, RO, SSA, cC1qR	19p13.13	2 mutations: type-1 = 52-bp deletion type-2 = 5-bp insertion In peripheral granulocytes and BFU-E (type-1 mutation). More commonly early in disease.	Rare or absent	[5,106,107,108,109]
**Other**		MPL (*W515L*)	MPL proto-oncogene, thrombopoietin receptor	C-MPL, CD110, MPLV, THCYT2, THPOR, TPOR	1p34.2	Most common acquired mutations: *W515L* *W515K* → JAK-STAT pathway constitutive activation. More commonly early in disease. Few cases of JAK2*V617F*-pos. PV & post-PV MF patients.	Rare	[5,106,110]

Abbreviations: MF = Myelofibrosis; neg. = negative; pos. = positive; PV = Polycythemia Vera; ↗ = means “increase”; →: indicates a causal relationship (e.g., “lead to…”).

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
