# Peer review of "Genetic Background of Polycythemia Vera"

_genes, 2022, doi:10.3390/genes13040637_

Round 1

Reviewer 1 Report

In their paper the authors present a comprehensive catalogue of genes that are mutated in polycythemia vera (PV) with detailed information about their function and prevalence. These genes are grouped together depending on their function, associated pathway and molecular mechanism. The review brings all this information together in one source which will be a valuable reference when reviewing the genetic landscape of PV by next generation sequencing.

  1. Using all the information provided about the genetic landscape of PV can the authors develop the paper further to provide insights such as
    1. Is there any associations between genes of the same of different pathways?
    2. Are there genes that are mutually exclusive?
  2. Is it possible for the authors for provide visual representation of the mutated genes associated with JAK2 either using a Venn diagram or a Circos plat?
  3. Would it be possible to summarise the overview of the genetic associations or absences in an additional section?

Reviewer 2 Report

The manuscript entitled “Genetic Background of Polycythemia Vera” is a review article on the genetic mutations in Polycythemia Vera. The goal of the current review is to collect the available literature regarding the canonical and non-canonical mutations together with their clinical significance. This article provides an useful information for clinical hematologists. Anyway, I think that the Authors should add a paragraph about the molecular analysis describing the standardized techniques and more sensitive detection methods and comment them.
